# Analysis of Imperfect Rephasing in Photon Echo-Based Quantum Memories

**DOI:** 10.3390/e24101367

**Published:** 2022-09-27

**Authors:** Byoung S. Ham

**Affiliations:** Center for Photon Information Processing, School of Electrical Engineering and Computer Science, Gwangju Institute of Science and Technology, 123 Chumdangwagi-ro, Buk-gu, Gwangju 61005, Korea; bham@gist.ac.kr; Tel.: +82-62-715-3502

**Keywords:** quantum memories, photon echoes, quantum coherence control, retrieval efficiency

## Abstract

Over the last two decades, quantum memories have been intensively studied for potential applications of quantum repeaters in quantum networks. Various protocols have also been developed. To satisfy no noise echoes caused by spontaneous emission processes, a conventional two-pulse photon-echo scheme has been modified. The resulting methods include double-rephasing, ac Stark, dc Stark, controlled echo, and atomic frequency comb methods. In these methods, the main purpose of modification is to remove any chance of a population residual on the excited state during the rephasing process. Here, we investigate a typical Gaussian rephasing pulse-based double-rephasing photon-echo scheme. For a complete understanding of the coherence leakage by the Gaussian pulse itself, ensemble atoms are thoroughly investigated for all temporal components of the Gaussian pulse, whose maximum echo efficiency is 26% in amplitude, which is unacceptable for quantum memory applications.

## 1. Introduction

Quantum memories [1,2,3,4,5,6,7,8,9] have been intensively studied in the emerging field of quantum interfaces; however, they have recently faced serious challenges regarding node scalability [10,11,12] due to the short decoherence time of qubits. Qubit scalability plays an important role not only in quantum algorithms such as Shor’s prime number factorization [13] and Grover’s data search [14], but also for fault-tolerant quantum computers [15] and long-haul quantum communications via quantum repeaters [16]. For fault-tolerant quantum computing, both fidelity and retrieval efficiency play an important role in satisfying a minimum level of quantum error corrections [17]. For quantum communications, imperfect devices may allow potential eavesdropping via the low retrieval efficiency of quantum memories. Thus, both a long storage time and high retrieval efficiency of quantum memories become major parameters to determine the functionality of quantum technologies.

Over the last decades, photon echoes [18] have been intensively studied as a potential candidate for quantum memories owing to the inherent benefits of ultrafast, wide bandwidth, and their multi-mode information processing capabilities [1,2,3,4,5,6,7,8,9]. The key mechanism of photon echoes is the collective controllability of ensemble coherence. Collective controllability is achieved by an optical π pulse, resulting in the reversible coherence evolution of the ensemble. However, the optical π pulse induces a complete population inversion and causes inevitable quantum noises to the retrieved signals due to spontaneous and/or stimulated emissions. Unlike classical information processing, quantum information does not allow duplication (or cloning) of an unknown quantum state [19]. Thus, population inversion in photon echoes is a fundamental constraint that needs to be overcome for quantum memory applications [1,2,3,4,5,6,7,8,9].

A double-rephasing photon-echo (DRPE) scheme was adapted for quantum memories due to its inherent property of having no population residual in the excited state during the rephasing process [20,21,22,23,24,25]. Although the first echo in the DRPE scheme is treated as silent, so that it does not affect the second (final) echo [20,21,22], the inherent absorptive coherence of the second echo prohibits echo generations out of the medium [23]. To solve this absorptive echo problem in the DRPE scheme, a controlled double rephasing (CDR) echo protocol was proposed [1], where controlled coherence conversion (CCC) plays a key role for the ensemble phase control [23,24,25]. The CCC uses controlled Rabi flopping via population oscillation between the excited state in the DRPE scheme and a third state. Unlike a two-level system of photon echoes including DRPE, however, the three-level system of CDR gains a π phase shift for a complete Rabi flopping by CCC [24,26]. Such an optical phase gain was first observed and discussed in resonant Raman echoes [27] and was applied to photon echo-based quantum memories [1]. Thus, CCC has become a key mechanism with which to manipulate the absorptive coherence of DRPE. Recently, a DRPE without CCC was observed [20,21,22], despite this seeming to contradict CDR echo theory. Here, I thoroughly investigate the rephasing process by a typical Gaussian light pulse and identify that the observed DRPEs [20,21,22] are not contradictory to CDR echo theory but defects resulting from imperfect rephasing by the Gaussian light pulse.

## 2. Analysis: A Gaussian Pulse in Photon Echoes

Due to the Gaussian light profile G*_j_*, a commercial laser light pulse can simultaneously generate nearly all kinds of pulse areas ($\sum^{\;}\Phi_{j}$), where $\Phi_{j}=\int_{0}^{T}\Omega_{j}dt≅\Omega_{j}T$. Here, the Rabi frequency $\Omega_{j}$ only depends on G*_j_*, where *j* is the position along the transverse direction with respect to the pulse propagation direction, and T is the time duration of a square pulse, which can be easily obtained by modern electro-optic devices. To analyze the echo observations in DRPE schemes [20,21,22], which seemingly contradict CDR echo theory [23,24,25], ensemble coherence evolutions were numerically investigated for individual atoms interacting with the Gaussian distributed light. The axial distribution of the light pulse was excluded, because square pulses can be easily generated. To fully visualize the atom coherence evolutions, all decay rates of the medium were set to zero. Such an assumption is also practically correct to maintain a near perfect echo efficiency. Thus, the phase evolution of the ensemble coherence relies on both optical inhomogeneous broadening of the ensemble and nonuniform spatial distribution of the Gaussian light. For analytical simplicity, the analysis of nonlinear echoes [27,28,29] was not considered, mainly due to the practical limit of the single photon level of the input signal for quantum memories.

For the present analysis, time-dependent density matrix equations were numerically solved for a two-level ensemble medium interacting with resonant optical pulses in the Heisenberg picture under rotating wave approximations [30]: $\frac{d\rho }{dt}=\frac{i}{\hbar }\left[H,\rho \right]-\frac{1}{2}\left\{\Gamma ,\rho \right\}$, where *ρ* is a density matrix element, H is the interaction Hamiltonian, and Γ is a decay parameter. For this, a commercial laser light, whose spatial profile is Gaussian distributed, was taken as a light source. The Gaussian light induced a Gaussian beam profile of interacting atoms, where the atom distribution was divided into 41 subgroups in the numerical calculations to cover 99.55% of the total distribution. Each subdivision of the Gaussian light-interacting atoms was divided into 281 spectral groups at a 10 kHz spacing for 1.2 MHz (FWHM) inhomogeneous broadening. Those 281 spectral groups were individually calculated for the time-dependent density matrix equations and summed for all spatial components of the Gaussian distributed atom groups, unless otherwise specified. The Gaussian light-corresponding atom groups did not overlap (or interact) with each other, resulting in no interference among them due to transverse (spatial) distribution. The same was true for the photon-echo components due to the phase matching condition: $k_{echo}=2k_{rephasing}-k_{data}$. Thus, the sum of all 41 Gaussian-distributed atom groups denotes the overall ensemble coherence evolutions. In the DRPE scheme, only the emissive components in the second photon echo contribute to the echo observations.

The optical pulse duration was set to 0.1 μs, and the time increment in the calculations was also 0.1 μs. Initially all atoms were in the ground state |1〉 ($\rho_{11}=1$), and thus all initial coherences were $\rho_{ij}=0$, where i≠j. The present numerical calculations are time-interval independent, so that there is no accumulated error dependent upon the time interval settings. For a π pulse area, the corresponding Rabi frequency of the 0.1 μs–pulse was 5 MHz multiplied by 2π. From now on, the 2π multiplication factor is omitted for simplicity. Here, the pulse area of a Gaussian light pulse in the transverse mode can be described with the Gaussian distribution G*_j_*. Unlike the Maxwell–Bloch approach [1,2,3,4,5,6,7,8], the present numerical calculations can show the details of coherence evolutions in the time domain without approximations. The Maxwell–Block approach of the same model has already been discussed [31]. This is the essential benefit of the present numerical method for photon-echo analyses.

Figure 1 shows the conventional two-pulse photon-echo simulation results, where the light pulse(s) has a Gaussian profile in a transverse direction, as mentioned above. The spatial magnitude of the light intensity perfectly maps onto the cross section of an interacting ensemble, resulting in a Gaussian distribution of atom groups. For simplicity of symmetry, we only took the one-dimensional transverse mode of the cross section. For the analysis of the present Gaussian pulse-based photon echoes, three types of two-pulse combinations can be considered: The first column in Figure 1 is for a Gaussian pulse applied to the data (D) only; the second column is for a Gaussian pulse applied to the rephasing (R) only; and the third column is for Gaussian pulses applied to both D and R. The first row of Figure 1 shows the three different cases of pulse combinations. The second row shows the corresponding density matrix calculations of the first row, where the complete rephasing appears at t = 4.1 μs (dotted line) for an echo. The third row shows the peak amplitude (red open circles) of the photon echoes in the second row along the dotted line, where the solid line represents the theory of the best-fit curve.

For the first case of the Gaussian D in the first column of Figure 1, the echo in the second and third rows perfectly fits the function of *sin(G_j_)*, where G*_j_* represents the Gaussian profile. This exactly mimics the area theorem of photon echoes [32,33], where an optical π pulse perfectly rephases all of the data components. Although the area theorem originates in the time domain, the echo efficiency reaches 100%, as shown in the lower left corner, where the echo efficiency η (square root of retrieval efficiency) is defined by the coherence ratio. Here, the D-pulse-induced coherence profile perfectly overlaps with the echo profile. Regarding the signal D in Figure 1, a typical two-pulse photon-echo scheme was adapted for visualization purposes. For practical quantum memories, however, the single signal D is considered as a single (few) photon. The spectral broadening of D is given by the light source, where the inhomogeneous broadening of the optical medium must be broader than that of D to satisfy the photon-echo scheme. A detailed photon-echo analysis in an optically dense medium was also studied [34,35].

For the second case of the Gaussian R in the second column of Figure 1, the theoretical best-fit curve *sin^2^(G_j_)* was obtained from the data. Because the π rephasing pulse is considered as a double coherence excitation of D, the square law is quite reasonable. Here, all components of echo efficiency are lower than unity except for the π/2−π pulse sequence at line center of G_j_. The resulting echo efficiency is cut by ~50%. In fact, the echo efficiency can be increased by expanding the width of the Gaussian profile up to 100%, in which 100% echo efficiency corresponds to a uniform rephasing pulse, as shown in the first column.

For the last case of both Gaussian pulses in the third column of Figure 1, the echo distribution profile is shrunken to a narrower extent than that of the coherence excited by the Gaussian D pulse, where the echo efficiency reaches ~70%. Interestingly, the best-fit curve is *sin^3^(G_j_)*, which is the product of the first and second cases. Here, the Gaussian pulse can be interpreted to induce a 30% coherence leakage due to imperfect rephasing process by the Gaussian rephasing pulse. This coherence leakage in the rephasing process allows unpermitted echo generations in the DRPE of refs. [20,21,22] (as discussed in Figure 2 and Figure 3). From the analyses in Figure 1, we can see the Gaussian rephasing pulse results in imperfect rephasing process-caused coherence leakage.

## 3. Results

Now let us consider the unpermitted DRPEs observed recently, where the first echo is erased macroscopically [20,21,22]. Figure 2a shows a typical pulse sequence of the DRPE scheme composed of identical double rephasing pulses. Here, the first echo generation is assumed to be silent, where this silencing process does not affect individual coherence evolutions [20]. As discussed in refs. [23,24,25,26], the second echo generation out of the medium in DRPE is strongly prohibited due to its absorptive coherence, if spatially uniform optical π pulses are used. Then, the DRPE observations must be due to emissive components, resulting from the imperfect rephasing process in Figure 1. Figure 2b shows a special case of the linearly varying rephasing pulse profile of R1 and R2, where this variation is spatially distributed in a transverse mode. This spatial variation of the rephasing pulses R1 and R2 corresponds to the pulse area varying from zero to 2π (10 MHz in Rabi frequency) for a fixed π/2 pulse area of D.

Figure 2c,d represent corresponding numerical calculations for Figure 2a,b, where emissive components of E2 are generated when the rephasing pulse area falls below ~π/2 or above ~3π/2. This is obviously due to the imperfect rephasing by R1, as discussed in Figure 1. Figure 2e is the plot of echo amplitudes of Figure 2c at each echo timing as a function of the rephasing pulse area. The echo E2 (red open circles) has certain regions of emissive components (Imρ12>0). These emissive components of E2 are due to imperfect rephasing in E1, where the rephasing pulse area Φ*_R_* satisfies the following inequality:(1)$$n<\frac{\Phi_{R}}{\pi }<n+\alpha$$
where α is 5/8, and n = 0, 1, 2,… The Φ*_R_* equally applies to both R1 and R2. Each solid curve represents the best-fit curve for the data in Figure 2c, where the best-fit curves (echo amplitudes) for E1 and E2 are intuitively obtained as
(2)$$E1=sin^{2}\left(\frac{\Phi_{R}}{2}\right)/2$$
(3)$$E2=-\sqrt{2}\left[sin^{2}\left(\frac{\Phi_{R}}{2}\right)[0.3-cos^{2}\left(\frac{\Phi_{R}}{2}\right)\right]$$

The first echo E1 is similar to the second case (second column) in Figure 1. Here, the condition for the silent echo E1 is satisfied without violating the generality of the present analysis, because the silence does not alter the individual coherence evolutions [20]. Thus, we conclude that the final echo E2 in the DRPE scheme can be observed if the rephasing pulse area Φ_R_ satisfies Equation (1). This condition is theorized by Equation (3) for E2>0. As a result, the echo observations in the DRPE experiments of refs. [20,21,22] are due to imperfect rephasing by Gaussian light pulses, without contradicting CDR echo theory [24]. Figure 2f shows that all components of the final echo E2 overlapped in the same time domain.

Figure 3 shows the case of π−π pulse sequence of DRPE, where all the light pulses interacting with a medium are Gaussian distributed along the transverse direction with respect to the beam propagation direction. The Gaussian spatial distribution of the light in Figure 3a is divided into 41 groups, where each group interacts with independent atoms whose inhomogeneous broadening is 1.2 MHz (FWHM). The data pulse is assumed to be weak: $\Phi_{D}=\pi /5$. The two red open circles in Figure 3a indicate the critical value calculated by Equation (3) for E2=0. Figure 3c,d show the results of the individual coherence evolutions for Figure 3b. In Figure 3d, the second echo E2 shows partially emissive components, satisfying Equation (1) (see the dotted circles). In Figure 3e, the first echo E1 at t = 6.1 μs, and the second echo E2 at t = 12.1 μs are shown with respect to the Gaussian distributed spatial mode. The red curve only represents the emissive components E2_eff_ ($Im\rho_{12}>0$ in E2) selected from the second echo E2 represented by the dashed curve. The emissive regions in E2_eff_ also satisfy Equation (1) for $\alpha ≅\frac{5\pi }{8}\;$. Here, the negative components of E2 do not alter the final echo generation E2 due to complete independence of the interacting atoms determined by the spatial distribution of the Gaussian pulse in Figure 3a. Figure 3f shows the sum coherence evolution for E2_eff_ in Figure 3e in the time domain. The echo efficiency η (E2_eff_/D) in Figure 3f is 6.9 %. For η, the peak amplitude of each coherence was compared, where the actual data-induced coherence is bigger than it appeared because Figure 3f only denotes $Im\rho_{12}>0$ at the second echo timing in Figure 3d. It should be emphasized that the first echo E1 was treated as not being macroscopically rephased [20] even if it appeared in the present calculations. Here, the sign of atomic coherence determines whether the coherence behaves as an absorptive or emissive signal [36].

Figure 3g shows the accumulations of Figure 3f for different peak pulse areas of the Gaussian rephasing pulse in Figure 3b, where the peak pulse area of the rephasing Gaussian profile varies from zero to 2π at a π/4 step, by increasing the rephasing Rabi frequency Ω_R_ (=Ω_R1_ = Ω_R2_). As indicated in Figure 3f, the data pulse-induced coherence appeared at different magnitudes because of the different amounts of emissive components for E2 for $Im\rho_{12}>0$. In Figure 3h, echo efficiency η $\left(\left|\frac{E2_{\mathrm{eff}}}{D}\right|\right)$ is plotted for different rephasing pulse areas in Figure 3g (see the dashed circle), where η shows a damped oscillation as the rephasing pulse area increases. The damping is due to the inhomogenously broadened atoms as discussed in ref. [25]. From Figure 3h, it is concluded that the unpermitted photon echo E2 in the DRPE scheme always exists regardless of the rephasing pulse area if the rephasing pulses are Gaussian distributed. The maximum echo efficiency of 26% is achieved when the rephasing pulse area is π/2 at Gaussian peak. The η converges at ~10% as the rephasing pulse area increases (see the best-fit curve). Figure 3 now clearly explains how unpermitted echo observations in the DRPE schemes (in refs. [20,21,22]) were possible, showing that it is a defect of Gaussian rephasing pulses. In other words, the unpermitted echo generation in the DRPE scheme of Figure 3b is understood as a coherence leakage due to the imperfect rephasing by Gaussian pulses. The low efficiency of DRPE as shown in Figure 3h, however, has no direct relationship with fidelity as long as spontaneous or stimulated emissions are not involved.

## 4. Conclusions

In conclusion, a rephasing process in a double-rephasing photon-echo (DRPE) scheme was thoroughly analyzed for ensemble coherence evolutions. The contradictory DRPE observations were identified to be a result of imperfect rephasing processes by the Gaussian light pulses. The theoretical formula for the unpermitted echo observations in DRPE was also theoretically induced. As a result, there were always positive echo generations regardless of the rephasing pulse area in the DRPE scheme if the rephasing pulse was Gaussian distributed. The resultant echo efficiency showed a damped oscillation with respect to the Gaussian rephasing peak-pulse area, where its maximum efficiency was 26% in amplitude. Considering potential eavesdropping in quantum channels due to device imperfection-caused loopholes [37] and errors accumulated in each gate operation for quantum error-corrections [17] in fault-tolerant quantum computing [15,38], achieving near perfect retrieval efficiency is an essential requirement for quantum memory implementations. For a near maximal echo efficiency, the CDR echo protocol must be satisfied with spatially uniform rephasing pulses.

## Figures and Tables

**Figure 1 entropy-24-01367-f001:**
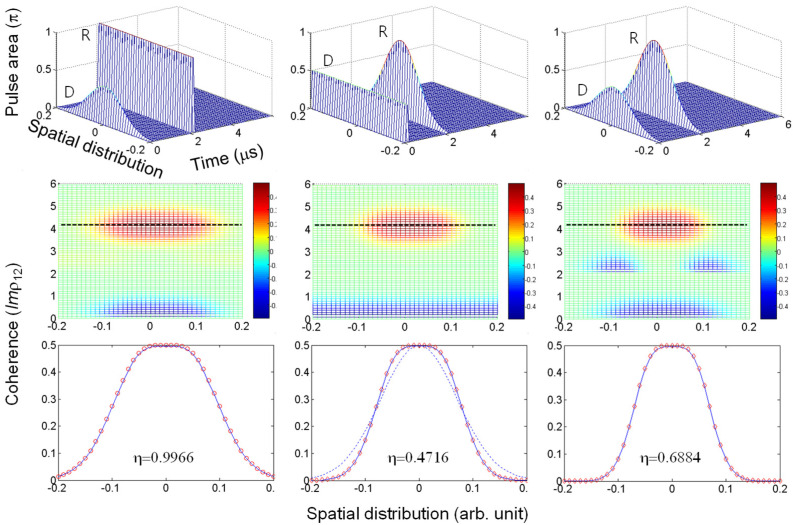
Gaussian pulse-dependent two-pulse photon echoes. Top row: three-different combinations of Gaussian pulse applications. Middle row: corresponding results of the top row. Bottom row: corresponding photon-echo efficiency η in amplitude. The arrival times of data (D) and rephasing (R) pulses are t_D_ = 0.1 μs and t_R_ = 2.1 μs, respectively. The Gaussian distribution is along the transverse direction of the beam propagation. The blue curve in the bottom raw indicates a theoretical fit, where the red circles are for the corresponding echo signals at t = 4.1 μs along the dotted line in the middle row. From left to right on the bottom row, the theoretical fit curves are *sin(G_j_)*, *sin^2^(G_j_)*, and *sin^3^(G_j_)*, where *G_j_* is the Gaussian distribution in the top row. Dotted curve: the Gaussian profile. The echo efficiency η is defined by $\frac{\left|\sum_{j}\rho_{12}\left(t_{e}\right)\right|}{\left|\sum_{j}\rho_{12}\left(t_{d}\right)\right|}$ [6,23,24,25,26], where *j* is the 41 groups in the Gaussian distribution, and *t_d_* (*t_e_*) is the data (echo) arrival time for maximum coherence.

**Figure 2 entropy-24-01367-f002:**
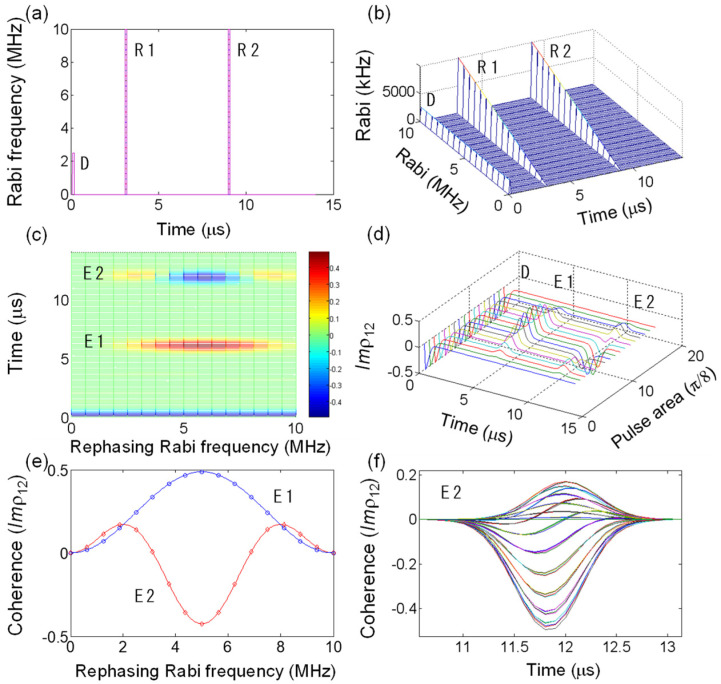
Doubly rephased photon echoes for linearly varying rephasing pulse area. (**a**) Pulse sequence for the double rephasing photon echo. D: Data pulse (π/2, fixed); R1: First rephasing pulse; R2: Second rephasing pulse. (**b**) Spatially varying rephasing Rabi frequency for (**a**). The 10 MHz Rabi frequency corresponds to 2π pulse area. (**c**) Corresponding results for (**b**). E1 (E2) represents the first (second) echo. (**d**) Individual atom phase evolutions for each varying Rabi frequency of the rephasing pulses in (**c**). (**e**) Rephasing Rabi frequency-dependent photon-echo amplitudes of E1 (blue circles) at t = 6.1 μs and E2 (red diamonds) at t = 12.1 μs: see Equations (1) and (2). Each colored curve is the best-fit theory (see the text). (**f**) Overlapped E2 in (**d**).

**Figure 3 entropy-24-01367-f003:**
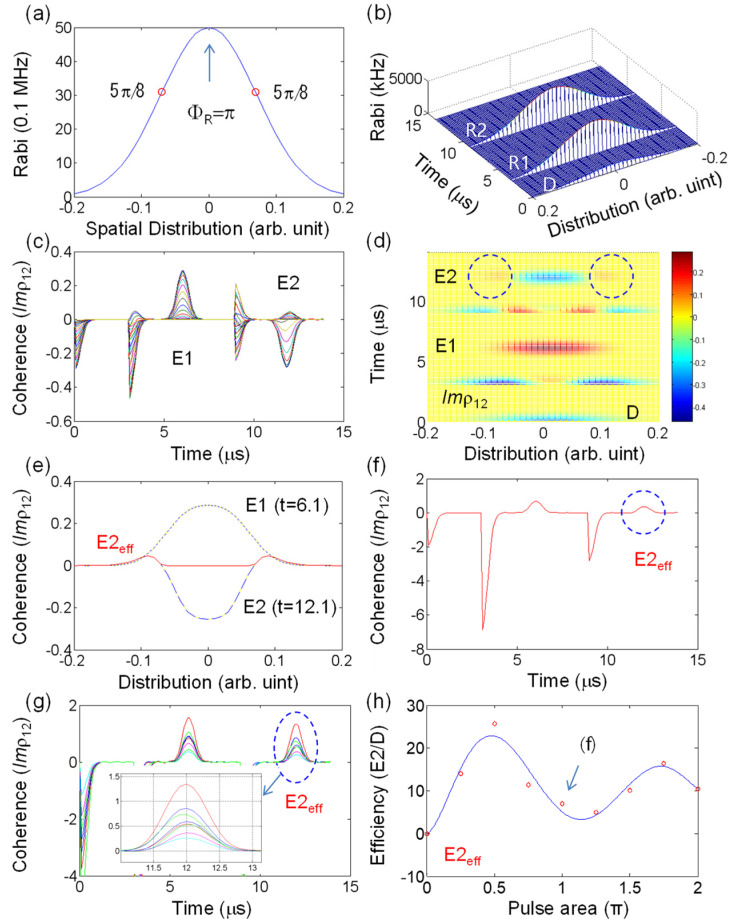
Gaussian pulse induced photon echoes in a double rephasing scheme. (**a**) Gaussian spatial profile of the rephasing pulse R1 and R2. The peak power (Rabi frequency) at center corresponds to a π pulse area. Red circles indicate the pulse area satisfying Equations (1) and (2). The Gaussian spatial profile of (**a**) is divided into 41 groups at 10 kHz spacing, as shown in (**b**). (**b**) Pulse sequence for a double rephasing photon echo. Data pulse area Φ_D_ is fixed at Φ_D_ = π/5. The arrival time of data D, rephasing R1, and R2 is 0.1, 3.1, and 9.1 μs, respectively. (**c**) Individual coherence evolutions of *Im*$\rho_{12}$ for all 41 components in (**b**). (**d**) A 3D color map for (**c**). (**e**) Coherence (*Im*$\rho_{12}$ ) as a function of Gaussian distribution for E1 and E2 in (**d**). The red curve is for E2_eff_, where $Im\rho_{12}>0$. (**f**) Sum of E2_eff_ for all Gaussian distribution in (**e**). (**g**) Overlapped sum coherence *Im*$\rho_{12}$ for different intensities of the Gaussian rephasing pulses in (b): for the peak pulse area (Φ_R_) in (**a**), Green: π/4; Red: π/2; Blue: 3π/4; Magenta: π; Cyan: 5π/4; Green: 3π/2; Blue: 7π/4; Red: 2π. The reason for a different magnitude in D-induced coherence is E2_eff_ (see the text). (**h**) Red circle: Echo efficiency η $\left(\left|\frac{E2_{\mathrm{eff}}}{D}\right|\right)$ in (**g**), where D stands for ρ_12_ (*t_d_*). Blue line: best-fit curve for the data (red circles). The arrow mark is for (**f**). The unit of coherence in (**f**,**g**) is arbitrary. The unit of efficiency in (**h**) is %.

## Data Availability

All results and data obtained can be found in open access publications.

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
