# Peer review of "Analysis of Imperfect Rephasing in Photon Echo-Based Quantum Memories"

_entropy, 2022, doi:10.3390/e24101367_

Round 1

Reviewer 1 Report

  The work is devoted to a detailed study of the formation of a photon echo when it is excited by resonant light pulses having a Gaussian profile in the beam cross-section. These issues are relevant in connection with the development of effective optical quantum memory, and the inhomogeneous distribution of the pulse area of exciting pulses in the transverse plane can greatly affect the degree of the echo signal retrieval and the efficiency of optical quantum memory, respectively. The author theoretically studies this issue in detail in an optically thin medium, in which the basic properties of photon echo are already laid. At the same time, the author considers various options for the excitation of primary echo and double-rephasing photon-echo (DRPE) known also as ROSE protocol [20], the latter is of great interest for the implementation of optical quantum memory based on it. As a result of the analysis, the author showed that Gaussian pulses can lead to the appearance of new unexpected signals. He also evaluated the efficiency of generating echo signals and determined the most optimal parameters of exciting pulses, allowing to achieve the greatest efficiency. It would still be possible to try to point out possible practical ways of using such pulses, when realizing the ultimate efficiency These results are of undoubted interest. In the final conclusion, the author inclines to the need to avoid using Gaussian pulses, which is fair.in the emission of echo signals.  

Given the overall value of the results obtained, I recommend the article for publication in the journal, taking into account a small number of technical comments, which are given below.

1.     1. When considering the issues presented by the author in the results of numerical calculations shown in Fig. 1, the author does not give the value of the pulse area of the first exciting pulse, it is desirable to indicate its value and if it is close to pi/2, then the influence of nonlinear effects on the shape and efficiency of the echo signal emission is already possible.

2.     2. Apparently, the author considers the exciting pulses to be extremely short, when it is possible not to take into account the frequency detuning of atoms during the action of the pulses. This is so, or, otherwise, it should be noted and compared with the corresponding parameters. Namely, it is desirable to compare the spectral width of the light pulses with the value of the inhomogeneous broadening of the line. The question is, how much is the line width smaller than the pulse spectrum? At the same time, I note that if we are talking about the implementation of quantum memory on a photonic echo, then the inhomogeneous broadening should be wider than the spectrum of the signal pulse [1]. And if in this case the spectra of the control pulses are comparable to the spectrum of the signal pulse, then taking into account atomic frequency detuning during interaction with light pulses becomes important [S.A.Moiseev, M.I. Noskov. The possibilities of the quantum memory realization for short pulses of light in the photon echo technique,  Laser Physics Letters, 1, â„– 6, 303-310 (2004)]. Here the formulas for calculating atomic coherence become much more complicated. This is not mentioned in the work, it is desirable to give some comments.

3.     3. The author evaluates the efficiency of echo emission in relation to the coherence excited by the signal pulses and the coherence  that causes the appearance of the echo signal (line 120-122). At the same time, it is generally accepted to evaluate the efficiency of quantum memory in relation to the energy of the echo signal to the intensity of the signal pulse. These values are also expressed in terms of signal intensities with the same time form. It is necessary to give an appropriate explanation about the motivation and advantages of the definition of effectiveness introduced by the author.

4.     4. The author compares the results obtained (see lines 124-127) for the two-pulse photon echo with the results of the area theorem of works [29,30]. In [30], an analytical solution was found for the pulse area not of the primary echo, but for the pulse area of all echo signals generated in an optically dense medium after the action of two exciting pulses. The solution can give an expression for the pulse area of a two-pulse echo signal in an optically thin medium, but this requires additional calculations. I note that an analytical solution for the pulse area of a two-pulse echo signal, valid even in an optically dense medium, was obtained in [S. A. Moiseev, Some general regularities of photon echo radiation in an optically dense medium. Opt. Spectrosc. 62, 180 (1987).]. A more detailed conclusion and analysis of the solution is given in recent works [R.V. Urmancheev, K.I. Gerasimov, M.M. Minnegaliev, T. Chaneliere, A.Louchet-Chauvet, S.A. Moiseev Two-pulse photon echo area theorem in an optically dense medium. Optics Express Vol. 27, No. 20, 28983 (2019); DOI: https://doi.org/10.1364/OE.27.028983;  S.A. Moiseev, M. Sabooni, and R.V. Urmancheev, Photon echoes in optically dense media, Phys.Rev.Research 2, 012026(R) (2020); DOI:https://doi.org/10.1103/PhysRevResearch.2.012026] .

5.     5.On the lines [215-216], the author introduces the total field E2eff. It is required to clarify in more detail how this quantity is related to atomic coherence ?

6.     6. On what basis is only the imaginary component of atomic coherence chosen during the emission of an echo signal? Since the echo signal is emitted in the absence of external coherent radiation, the question is with respect to which field component this polarization component is chosen and why?

Reviewer 2 Report

The efficiency of quantum memories is definetely a very important and interesting problem. However the relevance of the results presented in the manuscript are questionable. It seems that the authors avoid dynamical description of light since Maxwell Bloch equations are not solved. It is not clear whether such approach is appropriate for the problem. This conclusion can be wrong,  but then the authors should describe their model with more details. In the present version  it is even not clear what is the dimensionality of the problem.

Also it remains unclear  how the authors deal with decay channels. The master equation contains decay rates, but in the text the authors say that the rates are set to zero. Typical radiation decay for atomic medium is ~ 1e-8 s while the light pulse duration is 1e-7 s, thus the decay should be important for the atomic evolution and hardly can be ignored.

This makes me think that the presented model is oversimplified and should be extended.

In addition, there are multiple  phrases in the text that sound ambiguously. I'm not an expert in English but it seems to me that the phrase like  "individual ensemble atoms" should be modified. 

Reviewer 3 Report

The present manuscript reports on an analysis of rephasing mechanisms in quantum memories. In particular, the authors focuses on Gaussian light pulses for double-dephasing photon-echo scheme, to describe recent observations with such an approach.

Overall, I find the paper interesting and well written, and the presented analysis supports the conclusion. I am thus supporting its publication in Entropy. I would only suggest the author to fix a few remaining typos in the manuscript (e.g., "rephrasing" rather then "rephasing" in abstract).

Round 2

Reviewer 2 Report

I accept the authors reply and agree with the publication.